# Evacuation Simulation Implemented by ABM-BIM of Unity in Students' Dormitory Based on Delay Time

Yonghua Huang [1,2,3,4,*], Zhongyang Guo [1,2,3], Hao Chu [1,2,3,5] and Raja Sengupta [4]

1    Key Laboratory of Geographic Information Science, Ministry of Education, East China Normal University, Shanghai 200241, China
2    School of Geographic Sciences, East China Normal University, Shanghai 200241, China
3    Key Laboratory of Spatial-Temporal Big Data Analysis and Application of Natural Resources in Megacities, Ministry of Natural Resources, Shanghai 200241, China
4    Department of Geography, McGill University, Montreal, QC H3A2K6, Canada; raja.sengupta@mcgill.ca
5    Laboratory of Environmental Geosimulation (LEDGE), Department of Geography, University of Montreal, Montréal, QC H2V 0B3, Canada
*    Correspondence: yonghua.huang@mail.mcgill.ca

**Abstract:** China's university dormitories have high population densities, which can result in a large number of casualties because of crowding and stampedes during emergency evacuations. It is therefore important to plan properly for evacuations by mitigating the effect of choke points that create backlogs ahead of time. Accurate computer representations of the structure of a building and behavior of the evacuees are two important factors to obtain accurate evacuation time. In this paper, Agent-Based Modeling (ABM) and Building Information Modeling (BIM) are, respectively, implemented using the Unity platform to simulate the evacuation process. As a case study, the layout of a student dormitory building at Shanghai Normal University Xuhui District, Shanghai, China, is utilized along with the A* algorithm in Unity to explore the impact of evacuation speed and delays in creating choke points. Compared with previous research, the innovation of this study lies in: (1) using Unity software to make simulation of the physical environment both realistic and easy to implement, demonstrating Unity can be a well-developed platform to implement ABM-BIM research that focuses on crowd evacuation. (2) Using these simulations to evaluate different degrees of congestion caused by varying evacuation speeds, thus providing information about possible issues relating to evacuation efforts. Using the results, several recommended measures can be generated to help improve evacuation efficiency.

**Keywords:** Unity; students' dormitory; emergency evacuation; delay time; Building Information Modeling; Agent-Based Modeling; evacuation simulation; GIS

## 1. Introduction

University students in China who live in dormitories face particular risks resulting from congested conditions, particularly during fires. If students cannot be evacuated from their rooms inside a crowded dormitory to safe areas in a timely manner, it could result in casualties and property loss [1]. For example, in 2007, 500 students were trapped when clothes being dry cleaned caught fire in the dormitory of Northeast Normal University [2]. In the dormitory fire accident of Shanghai Business School in 2008, four students jumped from a building to escape quickly, but lost their lives [3]. In analyzing these types of accidents in dormitories, it is noted that any delays in the evacuation process is an important factor that predetermines the numbers of casualties. Studies have found that in the process of evacuation, the time spent walking or running to escape from the original location to a safe area is very short. In fact, most of the time is spent in the delay before the evacuation [4,5]. Therefore, research on how best to minimize delay time in the evacuation process is necessary to enable better management and optimization measures so

emergency rescue processes can be streamlined. In order to lower the evacuation time, it is imperative to analyze delay patterns in real-life situations with students within the standardized structure of dormitories. Further, in any building fire protection design, it is generally necessary to evaluate the evacuation time of individuals and compare this time with the critical time when the fire develops to an extent that causes mortality [6,7]. Since it is unrealistic to analyze the evacuation process during an emergency, computational modeling methods have all been implemented to simulate an emergency evacuation from buildings.

In general, the emergency evacuation problem can be regarded initially as a traditional routing problem to find the shortest path to an exit. Researchers have proposed improvements to several classical routing algorithms to solve this problem, e.g., Depth-First-Search algorithm [8], Dynamic Programming algorithm [9], Dijkstra algorithm [10], Ant Colony Optimization algorithm [11] and A* Search Algorithm [12]. Among them, the A* Search Algorithm is the most widely applied route searching algorithm.

However, evacuation is a complex process since the behavior of evacuees will have a significant impact on the evacuation results. To deal with these complications, researchers have proposed various other algorithms. Helbing's Social Force Model [13] and Adler's Cellular Automaton Model consider evacuees as a whole to model group evacuation behavior [14], while Agent-Based Modeling (ABM) focuses on simulating the actions of individual evacuees [15].

ABM has been shown to be an effective way to study complex systems [16], where an "agent" is an object that takes specific action after receiving stimulation from the environment [15]. An ABM can simulate all types of emergencies with low cost and low risk, which is especially suitable for simulating human behavior [17]. For example, ABMs have been used to simulate evacuations from various hazards, including indoor fires [18], tsunamis [19], typhoons [20], bushfires [21] and earthquakes [22]. Several ABM softwares exist (e.g., Netlogo, RePAST) to efficiently construct such models.

BIM (Building Information Modeling) is a building information expression method, proposed by Eastman in 1974, that expands the planar representation of a building's characteristics from a two-dimensional layout plan to three-dimensional space [23]. Since a BIM can be implemented in an evacuation simulation, its use in design and construction has expanded rapidly in recent years [24]. In one study, evacuation efficiency was optimized by developing a BIM-based virtual environment [25], and another study assessed the building regulations using a BIM-based evaluation system [26]. In a more recent study, a BIM-based 3D students' dormitory model was built to implement ABM for evacuation simulation time [27].

There are two issues in the current ABM simulations of evacuation. First, most existing ABM software have limited integration with 3D visualization to create realistic environments, with GAMA being an exception [28]. Second, and related to the first issue, existing evacuation models focus on the routing algorithm without paying attention to the physical 3D simulation environment. This leads to a model that presents a rather mechanized evacuation behavior while ignoring other behaviors (e.g., pre-evacuation time). Therefore, the motivation of this study is to provide a more reasonable virtual 3D environment for evacuation simulation research while considering other factors., i.e., pre-evacuation delays, when a real emergency occurs. The aim of this study is to verify that Unity [29] is a feasible ABM-BIM evacuation simulation platform that realistically simulates the 3D environment as well as the behavior of individuals as agents who traverse this environment.

## 2. Methodology

### 2.1. ABM and BIM in Unity

Currently, two types of evacuation simulation software are widely implemented for evacuation simulation and disaster prevention: commercial software (e.g., STEPS, Pathfinder) and open-source software (e.g., such as NetLogo, Repast) [30]. Liang et al. [31] used STEPS

software to process an evacuation experiment on electric buses and Chu et al. [32] utilized Pathfinder to propose a framework of evacuation simulation for optimizing emergency management in urban residential communities. Ionescu et al. [33] used NetLogo software to simulate evacuations from educational institutions. Cimellaro et al. [34] utilized Repast software to simulate human behavior under an earthquake disaster. Thus, it is apparent that software approaches, both commercial as well as open source, can add important insights into the evacuation process.

However, the above-mentioned software also have limitations: both the commercial software mentioned above, STEPS and Pathfinder, provide a 3D modelling environment but limit the users to the embedded evacuation rules, thus not giving them enough freedom to expand the evacuation functions [35]. As freeware, Netlogo and Repast support the programming of specific behaviors, but they can only implement an evacuation simulation in a 2D environment; thus, the evacuation process simulated is not completely consistent with reality [32]. In comparison, Unity is rarely used in evacuation studies. However, Unity has its own advantages: (1) Compared with commercial evacuation simulation software, Unity integrates a routing algorithm based on A* search algorithm [36] and supports a programming language interface to customize functions. Thus, every single evacuee can be conceptualized as an agent who can be evacuated according to the routing algorithm. (2) Compared with open-source evacuation simulation software, Unity can provide a full 3D modeling environment and realistic physical collision effect. Further, a three-dimensional model of the building (akin to a BIM) can also be easily created, a benefit of using a game-development engine.

Unity can also bring these advantages into evacuation research. Fei [37] has conducted evacuation flow research in multiple exit architecture on Unity to find out shorter evacuation times with more exits. However, this study did not take into consideration a pedestrian's delay time. Rahouti [38] demonstrated that Unity could be useful tools to develop assisted evacuations, but used another software for 3D modeling, thereby increasing the complexity of their study. Stigall [39] implemented an evacuation model in Unity that could be viewed using Microsoft HoloLens, but did not provide any statistics nor carry out an analysis of their results.

One of the main advantages of Unity is that its navigation function uses the A* search algorithm, along with a triangulated navigation mesh to generate paths (Figure 1), which is represented as a series of points on edges of mesh triangles.

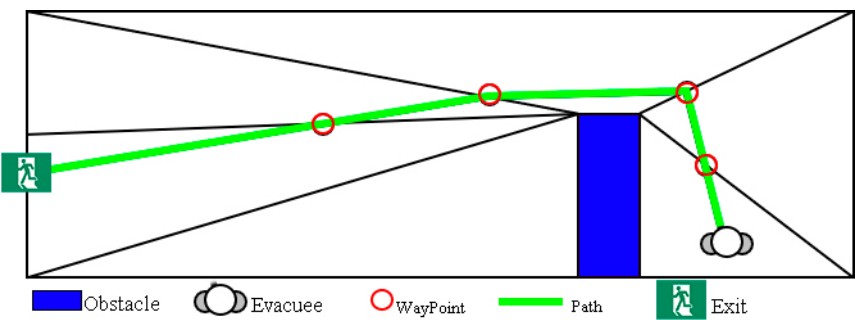

**Figure 1.** Path generation in A* search algorithm.

Figure 1 shows the projected path of an evacuee in a rectangular room. The evacuee is standing on the right side and plans to exit out on the left side. The navigation mesh is shown by the thin lines; an obstacle prevents the evacuee from walking straight to the exit. The path of the evacuee is shown as the green line and the waypoints are shown as red circles. A waypoint is generated for each edge that intersects the path.

In our study, Probuilder was used to implement the BIM part. Probuilder is a build-in tool for Unity to establish 3D models conveniently and includes stairs, planes, walls and corridors. Furthermore, models from Probuilder can be implemented with navigation mesh directly, as Figure 2 shows:

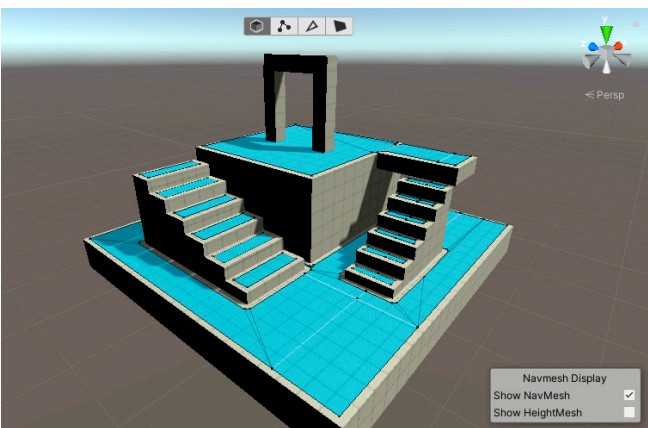

**Figure 2.** Example of the 3D model established by Probuilder.

In addition, Unity can give a realistic "movement effect" to evacuees, wherein the speed of the simulated evacuees will gradually accelerate from zero after starting the simulation and will gradually decelerate to zero when approaching the exit (instead of a sudden speed change). Further, evacuees will also have realistic collision effects, without simply passing through each other.

Based on the above considerations (i.e., the A* algorithm, realistic movements and collision effects), this study uses Unity software to determine the specific impact of "delay time" on the evacuation simulation results from dormitory buildings by combining BIM and ABM. It subsequently analyzes the simulation results. In this study, the simulated disasters are fires in a student's dormitory.

### 2.2. The Unity Game Engine

In this study, the Unity game engine was selected as the implementation platform to build the evacuation environment. A "game engine" refers to a set of components or systems that are pre-developed to allow developers to build the framework and core of a computer or video game easily. Using a game engine, developers can complete game production more quickly and efficiently through the functions and resources provided by the engine, rather than starting from scratch. At present, the two most commonly used game engines in the game industry are Unity and Unreal Engine 4. Among them, Unity is also widely used in the field of scientific research [40]. This study also uses Unity engine to construct the evacuation space. The Unity engine has powerful physical rendering effects, extremely low development costs and easy-to-use features, thus becoming the first choice for personnel simulation platforms [41]. The characteristics and advantages of the Unity engine are briefly introduced below.

#### 2.2.1. Basic Concepts and Hierarchical Relationship of Unity

The basic concepts of Unity include Scene, GameObject and Component. A completed game project contains several scenes; each scene is composed of several game objects and each object can mount several components.

#### 2.2.2. Operation Panels of Unity

The Unity editor has five panels: the Project panel, the Game panel, the Scene panel, the Hierarchy panel, and the Inspector panel. They can all correspond to panels with similar functions in ArcMap, as shown as Table 1.

**Table 1.** Function introduction and similar panels in ArcMap of Panels in Unity.

| Panels in Unity | Function | Similar Panels in ArcMap |
|---|---|---|
| Project | Storage whole data of one project | Catalog |
| Hierarchy | List the data currently needed | Table of Contents |
| Scene | Show all objects on Hierarchy panel | Layout View |
| Game | Show player's perspective | Data View |
| Inspector | show the properties of the selected object | Properties |

### 2.2.3. Advantages of Unity

As the evacuation simulation platform used in this research, Unity has the following advantages [42]:

(1) Unity has a simple and friendly operation interface, which is easy to operate, and various basic functions for making games and other projects have been realized and can be used directly. This allows for rapid development of the evacuation model.

(2) The products developed by Unity3D can be released on major mainstream platforms such as Windows, Mac, Linux, iOS, Android, etc. The switching of different platform versions only needs one-button deployment at the time of release without complex modification.

(3) Unity supports all mainstream file formats, taking picture files as an example, the formats include PSD, TIFF, JPG, TGA, PNG, GI, BMP, IFF, PICT, etc.; taking 3D model files as an example, the formats include FBX, DAE, 3DS, DXF, OBJ, MAX, MB, MA, etc.

(4) Unity currently supports two scripting languages, C# and Unity Script (JavaScript for Unity), with relatively low development difficulty.

(5) The hierarchical development environment of Unity makes the development work more efficient and orderly, more in line with the logical game development concept, and facilitates the control and modification of each object in the program.

(6) Unity supports and highly optimizes the graphics-rendering pipeline of DirectX and Open GL. Additionally, NVIDIA's PhysX physics engine is built in the software to provide physical effects such as rigid body, collision, joint motion and so on. Efficient graphics rendering and physical engine functions improve the authenticity of evacuation simulation.

## 3. Study Area and Evacuees

The simulated dormitory is the first floor of Yibei Building, Xuhui West Campus of Shanghai Normal University. The alarm system in the dormitory building uses a prerecorded voice broadcast system. The geographical location is shown in Figure 3, and the layout of the dormitory is shown in Figure 4.

The Yibei dormitory was selected for this research based on two reasons: (1) Lived experience by one of the co-authors, which added a familiarity with the internal environment of this dormitory building. (2) This dormitory building has been transformed from an office building, so the internal facilities are not completely suitable for dormitory living. In addition, each room is accommodated with 2–3 people, with the result that the internal space is relatively small, compared with other dormitory buildings. The density further necessitated creating an evacuation simulation for this building.

A single dormitory room of Shanghai Normal University on the first floor of the dormitory building is designed for accommodating four people (according to the information obtained from the dormitory Management Office) (Figure 5). This study simulates the situation where all the students are in the dormitory; however, their location in the dormitory is randomly placed.

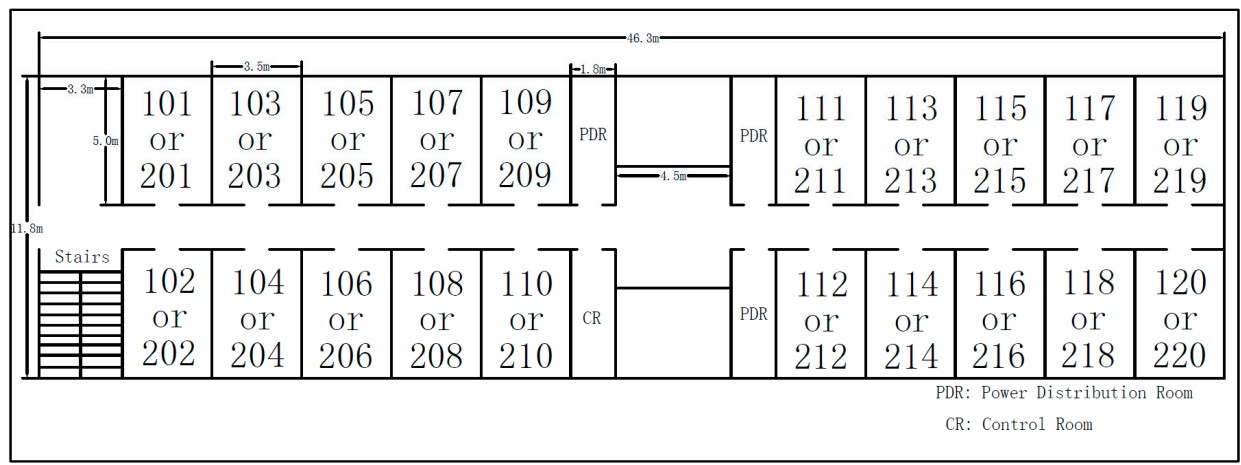

**Figure 3.** Study Dormitory in the Map of Shanghai Normal University.

**Figure 4.** Layout of Yibei Building.

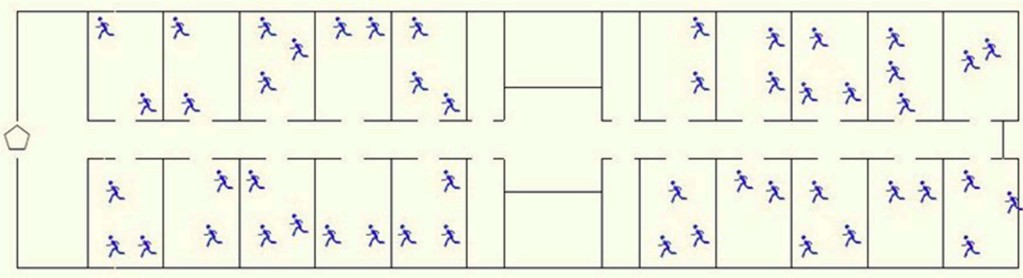

**Figure 5.** Distribution of personnel in Yibei building.

The evacuees simulated are college students, all around the age of 20, all male, with an average height of 1.7 m. Based on these statistics, the stride length of each person is about 50 cm, which refers to the distance between the centers of two feet after one step. People can walk about two steps in one second, and the speed will be faster when evacuating, but as many people will be evacuated in a single corridor at the same time, the speed is set at 1.5 m/s, 5 m/s and 10 m/s. These numbers reflect the range of speeds presented in the seminal paper by Helbing [43].

## 4. Delay Time before Evacuation

After a building fire occurs, whether people can safely evacuate depends on two characteristic variables: one is the available safe evacuation time or effective evacuation time (ASET), and the other is the required safe evacuation time (RSET) for the evacuees. It is generally considered that ASET > RSET is the deciding factor for the safe evacuation of the evacuees.

ASET is the critical time when the smoke develops to a condition threatening human life in a fire accident. RSET refers to the time interval from the moment of fire to the evacuation of evacuees to a safe area, which generally includes the following components:

$$t_{REST} = t_D + t_A + t_P + t_N \tag{1}$$

where $t_D$ is the time when the fire is detected (notification time); $t_A$ is the starting time of the automatic alarm system (reaction time); $t_P$ is the preparation time before evacuees' evacuation (preparation time); $t_N$ is the evacuation time of evacuees on the passage (walking time). Among them, the first three are called "delay time", namely:

$$t_{delay} = t_D + t_A + t_P \tag{2}$$

The distribution of specific evacuation time is shown in Figure 6:

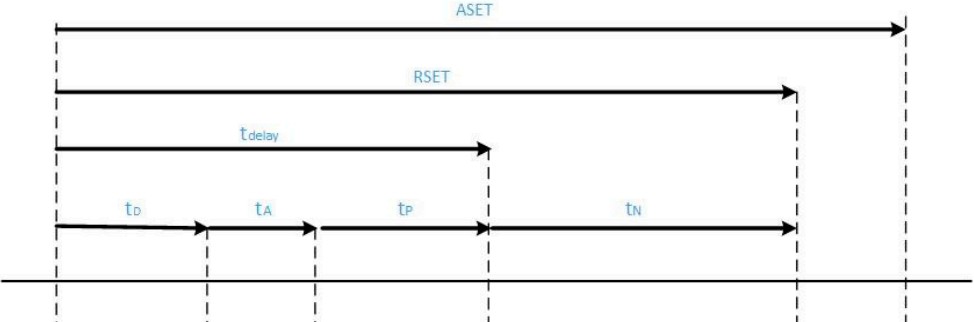

**Figure 6.** Distribution of evacuation time.

### 4.1. Notification Time

Notification time refers to the time from the occurrence of fire to the issuance of fire alarm signal, or when evacuees begin to feel the signs of fire. Signs of fire include seeing smoke, smelling smoke, seeing flames, breaking glass when hearing an alarm, a signal from a smoke alarm, activation of a water sprinkler system, etc. The notification time can be simulated or judged by professional knowledge. Table 2 shows the status of personnel in various locations under different alarm systems and the typical notification time of the alarm system [44].

**Table 2.** Statistical results of notification time when different alarm systems are used in buildings for various purposes.

| Location and Status of Personnel | Notification Time (min) | | |
| --- | --- | --- | --- |
| | Alarm System | | |
| | W1 | W2 | W3 |
| Office buildings, commercial buildings or industrial plants, schools (people are in a sober state, familiar with buildings, alarm systems and evacuation measures) | <1 | 3 | >4 |
| Shops, exhibition halls, museums, leisure centers, etc. (people are awake but unfamiliar with buildings, alarm systems and evacuation measures) | <2 | 3 | >6 |
| Apartments or boarding schools (people may be asleep but familiar with buildings, alarm systems and evacuation measures) | <2 | 4 | >5 |
| Hotels or hostels (people may be asleep and unfamiliar with buildings, alarm systems and evacuation measures) | <2 | 4 | >6 |
| Hospitals, nursing homes and other social public facilities (a considerable number of people need help) | <3 | 5 | >8 |

The type of alarm system in Table 1 is: W1-bit live broadcast instructions, using a sound broadcasting system, such as from the control room of closed-circuit television facilities; W2 is a non-live broadcast (pre-recorded) sound system, or visual information warning broadcast; W3 is an alarm system using alarm bells, siren or other similar alarm devices.

### 4.2. Reaction Time

Reaction time refers to the time from hearing the alarm or observing the signs of fire to deciding on action.

The actions that may be taken during the reaction time are: (1) the act of confirming the actual situation; (2) the act of protecting personal property; (3) the act of finding and gathering other family members; (4) finding and determining the appropriate exits and escape routes.

The length of the reaction time depends on: (1) the structural characteristics of the building; (2) the awareness and physical state of the personnel; (3) the type of alarm system, etc. The typical reaction time may be a few seconds (when the people are awake, well-trained and familiar with the building and alarm system) or a few minutes (when the people need help). Although a lot of work has been carried out to study people's reaction to the alarm system and the delay time used by individuals to prepare before evacuation, the reaction time is rarely mentioned in any evacuation simulation.

### 4.3. Preparation Time

Preparation time refers to the time when people prepare to evacuate or the time to find shelter areas. Pre-evacuation activities include all activities before deciding to leave or



actually starting to walk towards the exit or shelter area. These activities vary according to different building types. At present, there is no simulation technology in this regard. The analysis during this period depends on observation or expert judgment. In this study, reaction time and preparation time were considered together.

## 5. Evacuation Simulation and Results

The actual evacuation process of personnel is extremely complex and changeable. This study only conducts simulation experiments on "evacuees' delay time" as the research object. Therefore, before officially starting the evacuation simulation, it is clarified that:

(1)   The safe area of this evacuation simulation is the dormitory's outside door, that is, when evacuees arrive at the dormitory door that leads out of the dormitory, the evacuation is considered successful.

(2)   In addition to the delay time, the evacuees will definitely move to the safe area and will not stay in the evacuation space for a long time.

(3)   The evacuees will not be injured or killed due to crowding, collision and other factors during the evacuation. These are not simulated.

The detailed steps for establishing the ABM-BIM of evacuation as follows:

(a)   Using the Probuilder built-in extension within Unity to construct a 3D building model.
(b)   Creating a navigation mesh for the building space.
(c)   Adding a built-in auto-routing function for evacuees.
(d)   Programing the "delay time" feature and recording information feature for the evacuees.
(e)   Running the model to obtain the results.

### 5.1. Determination of the Evacuation Delay Time

As mentioned in the section above, the evacuation delay time is composed of the notification time, reaction time and preparation time. According to Table 2, the notification time is about 4 min. Considering that all the people living in the dormitory building are young men, the reaction time and preparation time were set to be at least 0 s and at most 90 s. Therefore, the evacuation delay time ranged from 240~330 s. In this study, the reaction time and preparation time were divided into three intervals: 0~30 s, 31~60 s and 61~90 s, and three simulations were performed for each interval. In addition, according to the "Codes of Fire Protection Design for Chinese Buildings" [45], the allowable time for the safe evacuation of general residential buildings need to be less than 6 min. Therefore, the available safe evacuation time ASET in this study was set to 360 s.

### 5.2. Establishment of the 3D Model of Evacuation Space

Before officially starting the evacuation simulation, it is necessary to model the evacuation scene. Figure 7 shows the three views of this dormitory building and Figure 8 shows the three-dimensional model of the dormitory built in Unity, which mainly includes four main parts: floor, room, stairs and exit.

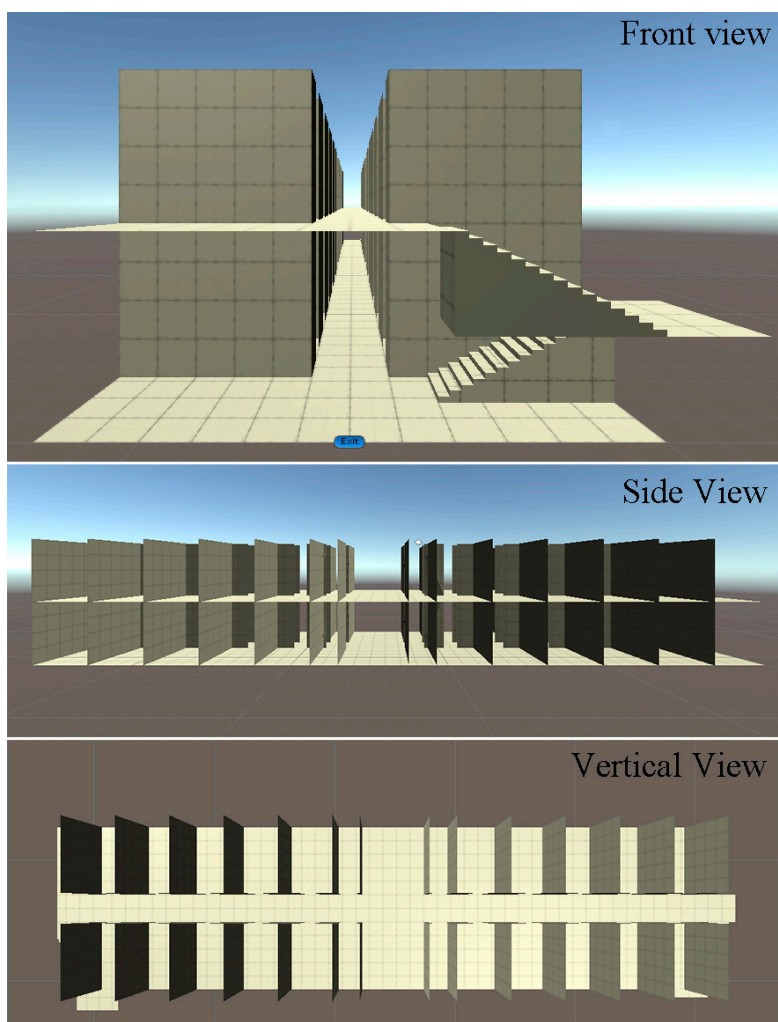

**Figure 7.** Three views of Yibei dormitory building.

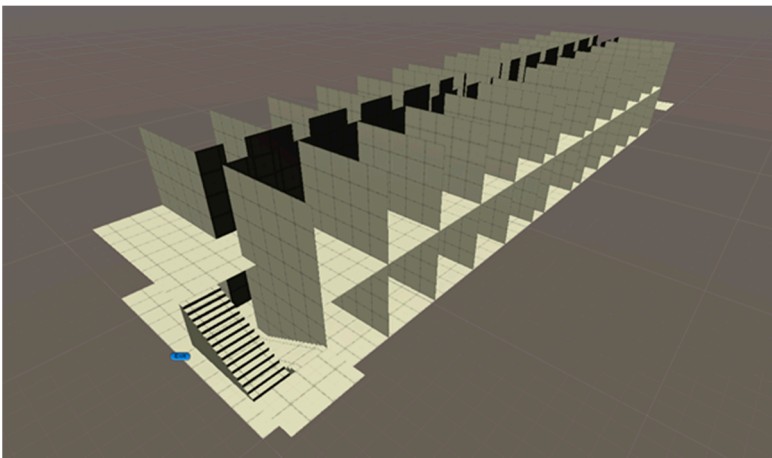

**Figure 8.** Three-dimensional model of the evacuation space.

*5.3. Evacuation Results*

In this study, the safe evacuation time and the easily congested area are taken as the simulation results.

### 5.3.1. Safe Evacuation Time

The safe evacuation time in the three simulations is shown in Table 3:

**Table 3.** Simulation results of the safe evacuation time under different evacuation delay times.

| Evacuation Delay Time Interval | Safe Evacuation Time RSET |
|---|---|
| 240~270 s | 305 s |
| 271~300 s | 337 s |
| 301~330 s | 369 s |

From the results of the safe evacuation time in Table 3, it can be seen that different evacuation delay times have a great influence on the safe evacuation time. When the evacuation delay time is between 301 and 330 s, the safe evacuation time is 369 s, which is greater than the previously estimated available safe evacuation time of 360 s. Based on this, it can be concluded that the evacuation delay time for people in an emergency should be controlled within 90 s.

### 5.3.2. Easily Congested Area

In this study, the spatial distribution of the cumulative real-time speeds was used to represent the easily congested areas in the study area, as shown in Figure 9. It can be seen from the figure that under different evacuation delay times, the speeds of the people are relatively low when they are on the first floor of the dormitory building near the staircase, the upper half of the stairway and the ground floor near the staircase, then these three areas are easily congested, with a drop in speed to as low as 0.25 m/s.

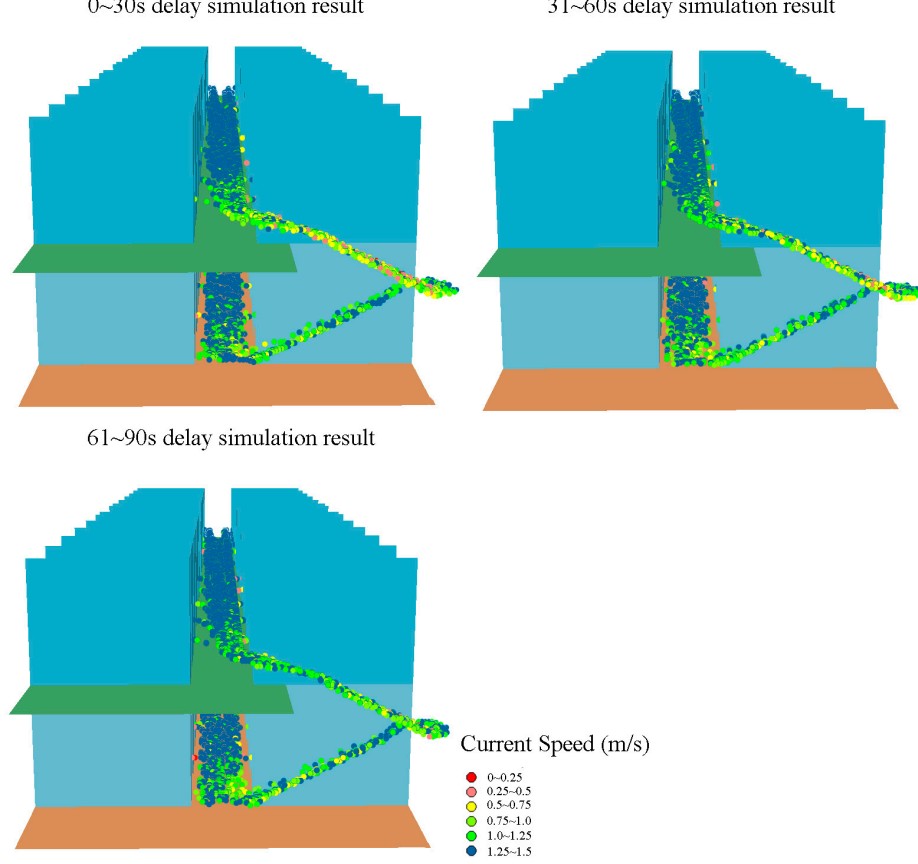

**Figure 9.** The spatial distribution of accumulated personnel real-time speed under different delay times.

5.3.3. Upper Limit of Evacuation Speed

The congestion caused by different evacuation speeds is also different, even in the same evacuation environment. Helbing [43] studied evacuation speeds from 1.5 m/s to 10 m/s and showed that it caused congestion, which led to deaths and stampedes when speed reach over 5 m/s. Similarly, in this study, 5 m/s and 10 m/s were therefore selected to test if the degree of congestion will increase (Figure 10).

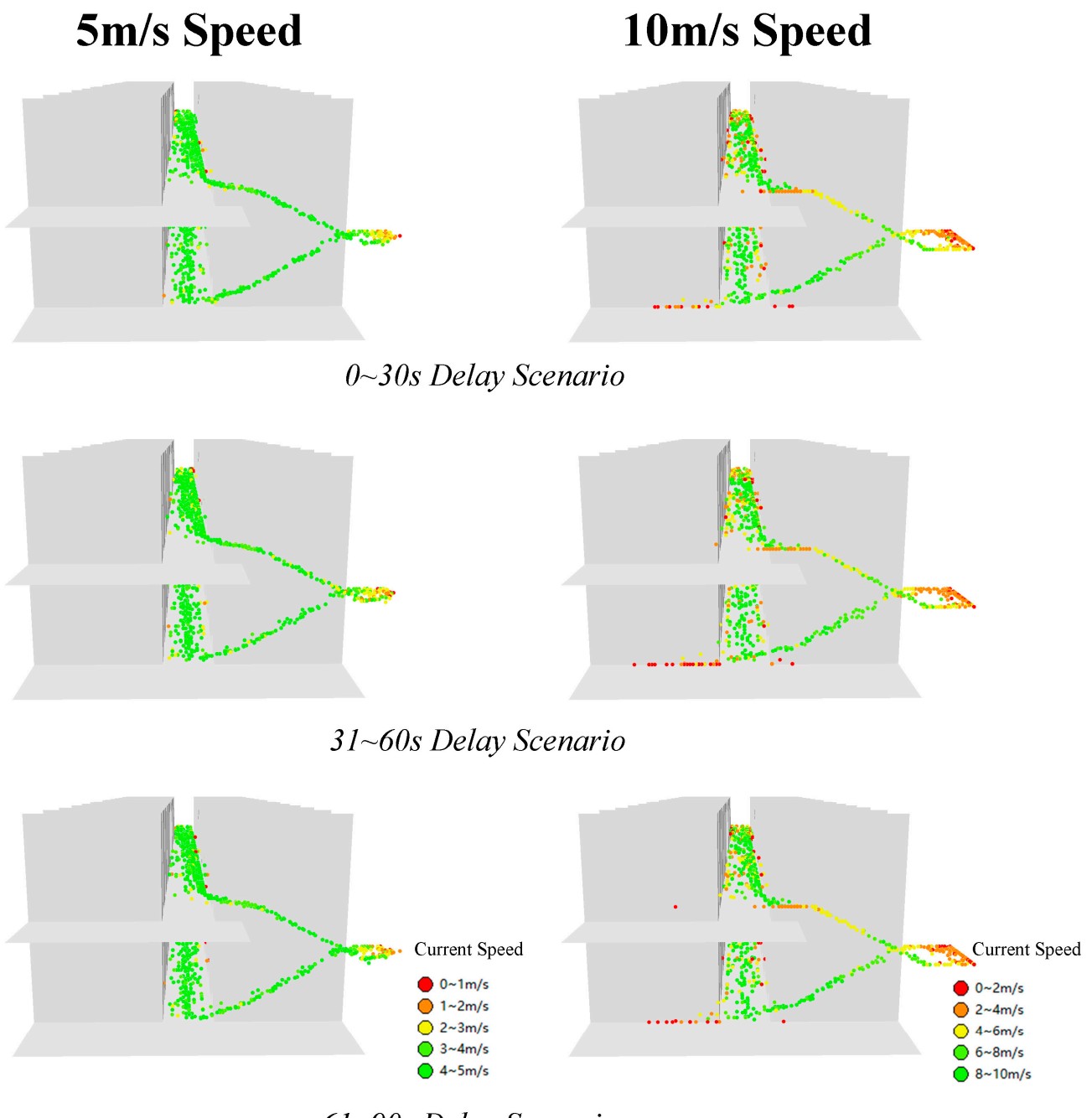

**Figure 10.** Comparison of congestion at different speeds under a different delay time scenario.

As shown in Figure 10, the evacuation congestion of the dormitory at the speed of 5 m/s is not that different from 1.5 m/s. However, at the speed of 10 m/s, the congestion of the dormitory is greatly enhanced. This is because at this high speed, the braking distance

of the evacuee is increased so that the simulated evacuee cannot stop at the exit, expanding the area of the area prone to congestion.

## 6. Conclusions and Future Prospects

Based on delay time, this research conducted a simulation evacuation study and analysis on the student dormitory building, and came to the following conclusions:

(1) From the results of the evacuation simulation time, it can be seen that the delay time has a greater impact on the total evacuation time. For this dormitory building, when the evacuation delay time of people is within 90 s, the evacuees in the building can complete the safe evacuation within the specified evacuation time.

(2) In this dormitory building, the first floor near the staircase, the upper half of the stairway and the ground floor near the stairway are areas that are easily congested. In practical application, the congestion in the above areas can be avoided by means of hierarchical fire broadcasting and evacuation guidance, so as to improve the overall evacuation efficiency.

(3) In the dormitory evacuation scenario, faster evacuation speeds (i.e., 10 m/s) may not lead to higher evacuation efficiency. On the contrary, it will probably cause even more serious congestion. Dormitory administrators should remind students not to panic and evacuate in an orderly fashion.

(4) The research on the delay time of crowd evacuation in emergencies is a relatively complex subject. Further research on this subject requires the establishment of a theoretical framework and computer simulation of individual behavior, as well as sociological behavior related to crowds from the perspective of psychology and sociology. This research conducts a more in-depth study on the delay time of crowd evacuation and draws some important conclusions. These conclusions will have a positive impact on the in-depth study of the subject and will provide important help for formulating reasonable crowd evacuation countermeasures.

In summary, there are two innovations in this study: one is exploring the impact of delay time during evacuation, another one is demonstrating that Unity can be a well-developed platform to implement an ABM-BIM research focus on crowd evacuation.

One of the limitations of the study is that the dormitory building referenced here has not conducted an evacuation drill. This is a necessary step before the model can be verified for accuracy, and to ensure the safety of all the residents.

Future research will expand the context of hazards to include natural disasters such as earthquakes, where other constraints emerge in evacuation plans of important buildings such as hospitals. In addition, similar research will be conducted in other parts of the world to factor varied social, economic and cultural characteristics.

**Author Contributions:** Conceptualization, writing–original draft, Yonghua Huang; project administration, Zhongyang Guo; formal analysis, investigation, Hao Chu; supervision, writing– review & editing, Raja Sengupta. All authors have read and agreed to the published version of the manuscript.

**Funding:** This research was funded by China Scholarship Council application number [202106140118]. And the APC was funded by Natural Science and Engineering Research Council of Canada [RGPIN-2022-04342].

**Institutional Review Board Statement:** Not applicable.

**Informed Consent Statement:** Not applicable.

**Data Availability Statement:** The authors confirm that the data supporting the findings of this study are available within the article.

**Conflicts of Interest:** The authors declare no conflict of interest.

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
