# Peer review of "Evacuation Simulation Implemented by ABM-BIM of Unity in Students’ Dormitory Based on Delay Time"

_ijgi, doi:10.3390/ijgi12040160_

Round 1

Reviewer 1 Report

In times when hazards including natural disasters strike communities, the importance of evacuation plans and activities becomes evident. I recommend conducting similar research considering other hazards especially natural hazards such as earthquakes and other types of buildings such as hospitals and considering other factors controlling the evacuation process. I also recommend conducting similar research in other parts of the world where different social and economic and cultural characteristics dominate.

Author Response

The authors thank the Reviewer for these excellent suggestions.

We have added details on the novelty of this work. In particular, we focus on the fact that ABM simulations often lack 3D perspective, whereas evacuation simulation software lack ability to model additional behavioural responses.

Further, for this manuscript, the authors prefer to focus on student dormitories, in order to draw attention to this important (and often under-discussed) issue. However the suggestions have been added to the Conclusions of the manuscript:

“Future research will expand the context of hazards to include natural disasters such as earthquakes, where other constraints emerge in evacuation plans of important buildings such as hospitals,. In addition, similar research will be conducted in other parts of the world to factor varied social, economic and cultural characteristics.”

Reviewer 2 Report

First of all, I would like to thank the authors for their effort and dedication in both the research and the presentation of their work.
The main point of innovation of this paper lies in the combination of ABM-BIM and Unity models, without exploring complex evacuation methodologies, only focusing on basic evacuation.
As a recommendation, a global edition in the presentation of the work is needed, as well as a better approach to the experiments, following a more robust methodological approach based on well-established theoretical foundations or rigorous data collection.
Throughout the article there are multiple unreferenced statements and data are taken for the modelling section that seem a priori arbitrarily selected.
Conversely, there are unnecessary figures (Figures 1 and 2) and very basic explanations of terms widely known in the area of knowledge that could be eliminated, simply quoting if necessary for the reader's information.
As for the conclusions, they are supported by the experiments conducted, but they are not very novel.

Author Response

(1) As a recommendation, a global edition in the presentation of the work is needed, as well as a better approach to the experiments, following a more robust methodological approach based on well-established theoretical foundations or rigorous data collection.

Response:.

We added one paragraph to state and provide a global edition in the presentation of Unity has been used to implemented evacuation researches:

“Unity can also bring these advantages into evacuation researches, Fei [37] has conducted evacuation flow research in multiple exit architecture on Unity to find out shorter evacuation time with more exits but lack of consideration for pedestrian’s delay time; Rahouti [38] demonstrated that Unity3D could be useful tools to develop assisted evacuation but completed 3D modeling by other software, increased the complexity of research; Stigall [39] did a good try to implement evacuation by using Microsoft HoloLens but lack of statistics and analysis of the results.”

[37] Lv, F. Research on simulation of pedestrian flow Unity 3D through multiple exit architecture. In Proceedings of the 2020 International Conference on Computer Engineering and Intelligent Control (ICCEIC), 2020; pp. 51-54.

[38] Rahouti, A.; Lovreglio, R.; Dias, C.; Datoussaïd, S. Simulating assisted evacuation using Unity3D. In Pro-ceedings of the Traffic and Granular Flow'17 12, 2019; pp. 265-275.

[39] Stigall, J.; Bodempudi, S. T.; Sharma, S.; Scribner, D.; Grynovicki, J.; Grazaitis, P. Building evacuation using microsoft HoloLens. In Proceedings of the 27th International Conference on Software Engineering and Data Engineering, 2018; pp. 8-10.

In order to provide more convincing basis of theoretical foundations, methodological approach and better experiments, four paragraphs and two pictures have been added to illustrate ABM-BIM theoretical basis of Unity in Chapter 2 “ABM and BIM in Unity” as follows:

“Navigation function in Unity uses the A* search algorithm and the triangulated navigation mesh to generate path (Figure 1), which is represented as a series of points on edges of mesh triangles.

Figure 1. Path generation in A* search algorithm

Figure 1 shows the projected path of an evacuee in a rectangular room. The evacuee is standing on the right side, plans to exit out on the left side. The navigation mesh is shown by the thin lines, an obstacle prevents evacuee from walking straight to the exit. The path of the evacuee is shown as the green line and the waypoints are shown as red circles. A waypoint is generated for each edge that intersects the path.

In this research, Probuilder was used to implement BIM part. Probuilder is a build-in tool for Unity to establish 3D models conveniently, includes stairs, planes, walls and corridors. Furthermore, models from Probuilder can be implemented with navigation mesh directly, as Figure 2 shows:

Figure 2. Example of 3D model established by Probuilder

In addition, developed physical environment of Unity can give real movement effect to evacuees, speed of evacuees will gradually accelerate from zero after starting simulation, and will gradually decelerate to zero when approaching the exit, instead of sudden speed change; evacuees will also have a real collision effect without passing through each other.”

(2) Throughout the article there are multiple unreferenced statements and data are taken for the modelling section that seem a priori arbitrarily selected.

Response: The data used to model the selected dormitory was collected in part based on lived experience by one of co-authors. In particular, the distribution of personnel in each room was obtained by visual observation. We added a paragraph to describe the source of data and explain the reason of selecting Yibei dormitory building in Chapter 4 “Study Area and evacuees “as follows:

 “The Yibei dormitory was selected for this research based on two reasons: (1) Lived experience by one of the co-authors, which added a familiarity with the internal environment of this dormitory building. (2). This dormitory building is transformed from an office building, so the internal facilities are not completely suitable for dormitory living. In addition, each room is accommodated with 2-3 people, with the result that the internal space is relatively small, compared with other dormitory buildings. The density further necessitated creating an evacuation simulation for this building.”

(3) Conversely, there are unnecessary figures (Figures 1 and 2) and very basic explanations of terms widely known in the area of knowledge that could be eliminated, simply quoting if necessary for the reader's information.

Response: To accommodate recommendations also by Reviewer 3 and Reviewer 4, Figure 1 has been replaced by Table 1.

Figure 2 (Figure 3 in current revision) has been retained in order to provide a clear geographical location of the research area.

(4) As for the conclusions, they are supported by the experiments conducted, but they are not very novel.

Response:

In order to highlight the unique contribution to current body of research, the following statement has been added to the conclusions: “There are two innovations in this study: one is exploring the impact of delay time during evacuation; another one is demonstrating Unity can be a well-developed platform to implement ABM-BIM research focus on crowd evacuation.”

Reviewer 3 Report

I was able to understand the entire flow of this paper easily. This paper especially has a good description of the motivation, background, methods, and importance of simulation evacuation research. I think that a few things need to be improved before it gets published, so here are my comments.

1)     Figure 6 does not clearly show me what the hallway looks like. It would be great if you could add another image showing the evacuation space from a different angle.

2)     Figure 7 visualization is not clear to me. It seems that they use proportional symbols to represent speed. But the circle size on the legend does not match the circle sizes of the images. I see that Figure 8 has the same issue. For example, the biggest red circle size in the legend looks much bigger than the red circle size in the image.

3)     I understand the importance of simulation evacuation research that the author conducted. But what are the author’s new contributions to the simulation evacuation research, and how is the uniqueness of this research compared to other similar research in this domain? I would suggest that these points should be highlighted in the section of the introduction or conclusion.

4) In the last section of section 2, the authors mentioned that this study uses Unity3D software to determine the specific impact of delay time on evacuation simulation results of dormitory buildings by combining BIM and ABM, then analyzes and verifies the simulation results. But I do not see how the authors verify the simulation results in the manuscript.

5) I rarely was able to see texts in Figure 1 because they are too small.

Author Response

(1) Figure 6 does not clearly show me what the hallway looks like. It would be great if you could add another image showing the evacuation space from a different angle.

Response: We have added Figure 7 to add clarity to the 3D space of the dormitory building.

“Figure 7 shows the three views of this dormitory building.

Figure 7. Three views of Yibei dormitory building”

(2) Figure 7 visualization is not clear to me. It seems that they use proportional symbols to represent speed. But the circle size on the legend does not match the circle sizes of the images. I see that Figure 8 has the same issue. For example, the biggest red circle size in the legend looks much bigger than the red circle size in the image.

Response: The evacuation results are presented in the 3D model, the size of the icon in the legend does not match that in the figure because of the effect of near large and far small. We have modified these figures to unify the size of the icon in the legend and distinguish different evacuation speeds only by color in Chapter 7.2 “Easily congested area”, as follows:

Figure 9. The spatial distribution of accumulated personnel real-time speed under different delay times.

Figure 10. Comparison of congestion at different speeds under different delay time scenario.

(3) I understand the importance of simulation evacuation research that the author conducted. But what are the author’s new contributions to the simulation evacuation research, and how is the uniqueness of this research compared to other similar research in this domain? I would suggest that these points should be highlighted in the section of the introduction or conclusion.

Response:

In order to highlight the unique contribution to current body of research, the following statement has been added to the conclusions: “There are two innovations in this study: one is exploring the impact of delay time during evacuation; another one is demonstrating Unity can be a well-developed platform to implement ABM-BIM research focus on crowd evacuation.”

(4) In the last section of section 2, the authors mentioned that this study uses Unity3D software to determine the specific impact of delay time on evacuation simulation results of dormitory buildings by combining BIM and ABM, then analyzes and verifies the simulation results. But I do not see how the authors verify the simulation results in the manuscript.

Response: Generally, the best way to verify the evacuation results is to conduct a drill. However given that a usually consumes a lot of manpower and material resources, we choose to simulate the evacuation. This has been stated as a limitation in the Conclusions:

“One of the limitations of the study is that the dormitory building referenced here has not conducted an evacuation drill. This is a necessary step before the model can be verified for accuracy, and also to ensure the safety of all the residents.”

(5) I rarely was able to see texts in Figure 1 because they are too small.

Response:

To accommodate recommendations also by Reviewer 2 and Reviewer 4, Figure 1 has been replaced by Table 1.

Reviewer 4 Report

Dear Authors,

In general, the topic of this paper sounds interesting and the proposed methodology is an innovative approach to be considered in order to model and solve evacuation routing problem. However, the link between the study and GIS application is not clearly mentioned or even exist in the content. Also, the motivation and aim of this research study is not clear to the readers. The problem statement and the gap in existing studies is not given. 

The following are my general comments:

1. The introduction section need to be revised and explain clearly the existing gap and the motivation of this study. The Aim should be revised as well. 

2. Some references are need in first paragraph of introduction section, line 53-56.

3. In Section 2, the ABM and BIM methods are not well described and not clear how these methods was used in this study. 

4. Section 3, Figure1. Better to display the results as graphs instead of screen short of the interface.

5. Figure 2. The map is not clear, need to be improved.

6. The implementation of ABM-BIM is no where described. It is good to give more details on implementation of these methods.

Author Response

(1). The introduction section need to be revised and explain clearly the existing gap and the motivation of this study. The Aim should be revised as well.

Response: We have added details on the novelty of this work. In particular, we focus on the fact that ABM simulations often lack 3D perspective, whereas evacuation simulation software lack ability to model additional behavioural responses.

We have added one paragraph in Chapter 1 “Introduction” to explain the existing gap, motivation and aim of this study as follows:

“There are two issues in current ABM simulations of evacuation.  First, most existing ABM software have limited integration with 3D visualization to create realistic environments, with GAMA being an exception [28].  Second, and related to the first issue, existing evacuation =, models focus on the = routing algorithm without paying attention to the physical 3D simulation environment. This leads to the model that presents a rather mechanized evacuation behavior while ignoring other behaviors (e.g., pre-evacuation time). Therefore, the motivation of this study is to provide a more reasonable virtual 3D environment for evacuation simulation research while considering other factors. i.e., pre-evacuation delays, when the real emergency occurs. The aim of this study is to (1) verify that Unity [29] is a feasible ABM-BIM evacuation simulation platform that realistically simulates the 3D environment as well as the behavior of individuals as agents who traverse this environment.”

(2). Some references are need in first paragraph of introduction section, line 53-56.

Response: We have added four references to Chapter 1 “Introduction” as follows:

“Since it’s unrealistic to analyze the evacuation process during an emergency, computational modeling methods such as Social Force Models [13], Cellular Automata [14], Agent-Based Models (ABM) [15] and Building Information Models (BIM) [23] have all been implemented to simulate the emergency evacuation of buildings.”

  1. Helbing, D.; Molnar, P. Social force model for pedestrian dynamics. Physical review E. 1995, 51, 4282.
  2. Blue, V. J.; Adler, J. L. Cellular automata microsimulation of bidirectional pedestrian flows. Transportation Research Record. 1999, 1678, 135-141.
  3. Poole, D. L.; Mackworth, A. K. Artificial Intelligence: foundations of computational agents; Cambridge University Press: 2010.
  4. Song, Y.; Wang, X.; Tan, Y.; Wu, P.; Sutrisna, M.; Cheng, J. C.; Hampson, K. Trends and opportunities of BIM-GIS integration in the architecture, engineering and construction industry: a review from a spatio-temporal statistical perspective. ISPRS International Journal of Geo-Information. 2017, 6, 397.

(3). In Section 2, the ABM and BIM methods are not well described and not clear how these methods was used in this study.

Response: We have added three paragraphs to Chapter 2 “ABM and BIM in Unity” to describe principle of ABM and how BIM was used in Unity, as follows:

“Navigation function in Unity uses the A* search algorithm and the triangulated navigation mesh to generate path (Figure 1), which is represented as a series of points on edges of mesh triangles.

Figure 1. Path generation in A* search algorithm

Figure 1 shows the projected path of an evacuee in a rectangular room. The evacuee is standing on the right side, plans to exit out on the left side. The navigation mesh is shown by the thin lines, an obstacle prevents evacuee from walking straight to the exit. The path of the evacuee is shown as the green line and the waypoints are shown as red circles. A waypoint is generated for each edge that intersects the path.

In this research, Probuilder was used to implement BIM part. Probuilder is a build-in tool for Unity to establish 3D models conveniently, includes stairs, planes, walls and corridors. Furthermore, models from Probuilder can be implemented with navigation mesh directly, as Figure 2 shows:

Figure 2. Example of 3D model established by Probuilder”

(4). Figure1. Better to display the results as graphs instead of screen short of the interface.

Response: To accommodate recommendations also by Reviewer 2 and Reviewer 3, Figure 1 has been replaced by Table 1

(5). Figure 2. The map is not clear, need to be improved.

Response: Thanks. Original Figure 2 has been replaced by currently Figure 3 with much higher resolution in Chapter 4 “Study Area and evacuees” as follows:

Figure 3. Study Dormitory in the Map of Shanghai Normal University

(6). The implementation of ABM-BIM is nowhere described. It is good to give more details on implementation of these methods.

Response: We have added detailed steps to implement ABM-BIM for evacuation in Unity in Chapter 6 “Evacuation and Simulation” as follows:

“The detailed steps for establishing ABM-BIM of evacuation as follows:

(a) Using Probuilder built-in unit to construct 3D building model.

(b) Creating navigation mesh on the building space.

(c) Adding built-in auto-routing function to evacuees.

(d) Secondary development and programing delay time feature and recording information feature to evacuees.

(e) Running the model to get the results.”

Round 2

Reviewer 2 Report

I would like to thank the authors for the changes made to clarify the various comments made by the reviewers and to considerably improve the quality and presentation of the results.

On the other hand, I would like to see a slight revision of the way the texts are written, paying special attention to the specific vocabulary used and the purely technical terminologies.

Author Response

Many thanks to the reviewer’s affirmation and suggestions on the revised manuscript. Some terminology has been modified here in the introduction in response to suggestions. For example, unify the software name as Unity, with the first letter capitalized and no suffix 3D. Because the definition of Unity on Wikipedia is a development engine that can develop 2D and 3D games.

At the same time, we have made many other wording changes in the article.

Reviewer 3 Report

The authors addressed issues.

Author Response

The authors thank the reviewer’s comments.

Reviewer 4 Report

Dear Authors,

The given comments are well addressed and the manuscript has been improved. However, there are still minor errors that need to be addressed such as text editing and the structure. I would suggest to merge some sections and facilitate readers to follow. For instance, use 4 to 5 sections or chapters (e.g., Introduction, methodology, study area,  Results and discussion, and conclusion).

Kind regards,

Author Response

Thanks to the reviewer’s affirmation and suggestions on the revised manuscript. We merged some chapters. Shortened the manuscript from 9 chapters to 6 chapters. See the figure below for details.
